# Are My Deep Learning Systems Fair?
# An Empirical Study of Fixed-Seed Training

**Shangshu Qian**
Purdue University
West Lafayette, IN, USA
shangshu@purdue.edu

**Hung Viet Pham**
University of Waterloo
Vector Institute
hvpham@uwaterloo.ca

**Thibaud Lutellier**
University of Waterloo
Waterloo, ON, Canada
tlutelli@uwaterloo.ca

**Zeou Hu**
University of Waterloo
Vector Institute
z97hu@uwaterloo.ca

**Jungwon Kim**
Purdue University
West Lafayette, IN, USA
kim3241@purdue.edu

**Lin Tan**
Purdue University
West Lafayette, IN, USA
lintan@purdue.edu

**Yaoliang Yu**
University of Waterloo
Vector Institute
yaoliang.yu@uwaterloo.ca

**Jiahao Chen**[*]
J. P. Morgan AI Research
New York, NY, USA
jiahao@getparity.ai

**Sameena Shah**
J. P. Morgan AI Research
New York, NY, USA
sameena.shah@jpmorgan.com

## Abstract

Deep learning (DL) systems have been gaining popularity in critical tasks such as credit evaluation and crime prediction. Such systems demand fairness. Recent work shows that DL software implementations introduce variance: identical DL training runs (i.e., identical network, data, configuration, software, and hardware) with a fixed seed produce different models. Such variance could make DL models and networks violate fairness compliance laws, resulting in negative social impact. In this paper, we conduct the first empirical study to quantify the impact of software implementation on the fairness and its variance of DL systems. Our study of 22 mitigation techniques and five baselines reveals up to 12.6% fairness variance across identical training runs with identical seeds. In addition, most debiasing algorithms have a negative impact on the model such as reducing model accuracy, increasing fairness variance, or increasing accuracy variance. Our literature survey shows that while fairness is gaining popularity in artificial intelligence (AI) related conferences, only 34.4% of the papers use multiple identical training runs to evaluate their approach, raising concerns about their results' validity. We call for better fairness evaluation and testing protocols to improve fairness and fairness variance of DL systems as well as DL research validity and reproducibility at large.

## 1 Introduction

DL systems, which consist of DL models and DL software (for example libraries such as Tensor-Flow) [71], are widely used; and enrusing their fairness is of paramount importance [47, 59, 97, 98]. Biases in DL systems have been identified as a major concern for civil rights by the U.S. government [27]. Yet, fairness testing for DL systems is still an open challenge.

Although the DL community is aware of the nondeterminism caused by using different random seeds [72, 90] , such algorithmic factors are not the sole source of DL nondeterminism. Even ***with the***

---

[*]Jiahao Chen has since moved to Parity Technologies.

*same seed*, e.g., the same initial weights, DL systems are nondeterministic, i.e., multiple training runs with the same network configuration, the same dataset, and the same seed, referred to as *fixed-seed identical training runs* (*FIT runs*), produce different models with different accuracies [71]. Such nondeterminism is caused by *software implementation*, e.g., software parallelism and floating-point computation of DL software (details in Appendix E.1).

Our previous work [71] shows that software implementation alone causes models' accuracies to vary by up to 2.9%, and most (83.8%) of the DL researchers and practitioners surveyed are unaware or unsure about the nondeterminism caused by software implementation. The small portion that is aware of such implemental variance underestimates it. The survey participants are researchers and practitioners with DL experience from universities and AI companies including Google, NVIDIA, Microsoft, and Facebook. These results suggest that even with fixed-seed identical training runs, one may still need to report error bars or use statistical tests.

Since the focus of the prior study [71] was on the variance of accuracy and training time only, it is unknown how DL software implementation affects the variance of *fairness*. Various debiasing algorithms [3, 47, 59, 98, 99] optimize DL models toward a specific fairness goal and it is unclear how software implementation affects these debiasing algorithms. In addition, unlike accuracy, which is a widely-accepted metric, dozens of fairness metrics exist [29, 35, 46, 60, 64, 99, 108]. It is unclear if software implementation causes different levels of variance for different fairness metrics.

In this paper, **we study the fairness variance caused by software implementation in training DL networks**. Overall, we found that fairness variance is large—up to 12.6% difference (27.3% versus 39.9%) across 16 fixed-seed identical training runs.

First, our results show that one training run may produce a fair model but another fixed-seed identical training run may generate an unfair one. For example, we reproduced previous work [47] that proposed a new network and a debiasing algorithm and we found that one FIT run generates a fair model (i.e., a model that does not discriminate based on gender) based on the threshold (20%) for the normalized disparate impact ($\overline{\text{DI}}$) metric [29, 64] ($\overline{\text{DI}}$ of 17.9%). A $\overline{\text{DI}}$ of 0% is the fairest, while a 100% $\overline{\text{DI}}$ is the least fair. However, a second FIT run produces an unfair model ($\overline{\text{DI}}$ of 20.4%, above the 20% threshold). This 20% threshold for $\overline{\text{DI}}$ has been used in *a U.S. legal case* to determine a system's fairness [9], where fairness variance could have changed the outcome.

Second, the variance of fairness imposes challenges for the valid evaluation of debiasing algorithms. For example, our experiments show that while a debiasing algorithm [107] reduces the bias amplification (BA) [99, 108] to 6.1% in one FIT run (compared with a BA score of 7.8% for its baseline), this technique *increases the bias amplification instead of reducing it* when we check the average of 16 FIT runs: an average BA score of 8.7% compared to an average BA score of 7.4% for the baseline. The Mann-Whitney U-test shows the results from the 16 runs are statistically significant. The observed variance across 16 FIT runs is solely caused by software implementation.

Finally, many mitigation techniques increase fairness variance or hurt accuracy. With Mann-Whitney U-test and Levene's test, our experiments show that a majority (68.0%) of the mitigation techniques reduce biases with a cost: they increase fairness variance, increase accuracy variance, or decrease model accuracy.

Since existing debiasing algorithms [47, 59] often do not measure the variance of biases, they could introduce risks. When such debiasing algorithms are applied to online learning, where DL models are trained continuously [53], the increased variance from debiasing algorithms could cause the resulting models' fairness to *miss* compliance requirements or service-level agreements. For example, Microsoft Tay [86] became a "racist chatbot" only after a few hours of deployment.

What's worse, even with fairness auditing before deployment, which is still *not* a standard practice in the industry [39], a model can be fair on pre-deployment data, but unfair on real-world data, due to the gap between training/test data and real-world data. The fairness variance puts the trained models at a greater risk of being unfair on real-world data. Theoretically, while debiasing techniques on average improve fairness on the training set, they increase the variance of fairness on the test set [2]. In practice, such gaps are the key reason that facial recognition systems [58, 76] have a racial bias.

In addition to quantifying fairness variance, **we conduct a survey of research papers** from top artificial intelligence (AI) conferences of recent years (2016–2020) to understand how researchers

handle the fairness variance caused by DL software implementations. Only 34.4% of the 32 papers evaluating or addressing DL fairness issues use multiple training runs in their evaluation.

To the best of our knowledge, we are the first to study software implementations' impact on DL fairness variance. The contributions are:

- A variance study of DL fairness of 27 techniques (22 mitigation techniques and five baselines). The experiments are performed on four popular datasets (CelebA, MS-COCO, imSitu, and CIFAR-10S) with three DL networks (ResNet-18, ResNet-50, and NIFR [47]), measured by seven popular bias metrics (Section 3).

  - **Finding 1**: Across 16 FIT runs, software implementation causes a maximum difference in bias metrics of up to 12.6% in the Demographic Parity (DP) metric and up to 11.8% in the $\overline{\text{DI}}$ metric. Yet, most (83.8%) researchers and practitioners are unaware or unsure about the DL nondeterminism caused by software implementation [71].
  - **Finding 2**: The effectiveness of the debiasing algorithms can be inaccurate with only one fixed-seed training run.
  - **Finding 3**: Most of (15 out of 22) the mitigation techniques tested increase at least one of the bias metrics.
  - **Finding 4**: In a majority (68.0%, or 70) of 103 experiments, debiasing algorithms have a negative impact on the model.

- A literature survey on the fairness research papers of top machine learning (ML), (NeurIPS/NIPS, ICML, AAAI, and FAccT), and computer vision (CVPR, ICCV, and ECCV) conferences (Section 4).

- Implications and suggestions for DL fairness researchers and practitioners (Section 5), including one should use FIT runs (setup details in Section 2.5) with statistical tests (Appendix B.3) to evaluate debiasing algorithms, when nondeterminism from DL software is unavoidable.

**Data and code availability**: Experiment data and artifact for reproducibility study are available in a public GitHub repository[1].

## 2    Experimental method

We investigate 27 techniques of DL fairness on four popular datasets with three networks by reproducing four previous works [47, 59, 98, 99] that fit our scope.

### 2.1    Scope

We focus on the impact of DL software implementation on the variance of group fairness [26] in classification tasks, where two orthogonal labels are assigned to each instance in the dataset, a *task label* for training a classifier, and a *protected label* that should not be discriminated against (e.g., gender). The set of instances with the same protected label forms a *protected group*.

Although the protected labels are not used in the classification task, the trained models could still be biased because the protected labels can be leaked into the prediction by proxy variables [3, 59, 98, 108].

### 2.2    Bias metrics

Various metrics exist to measure biases of ML algorithms. Yet, it is still an open problem to select these metrics as an optimization goal for real-world problems [49]. Therefore, we choose three fairness conditions and seven bias metrics derived from them. Fairness conditions include statistical parity [26] (SP), predictive equality [19] (PE), and equality of opportunity [35] (EOp). Seven bias metrics include demographic parity [15] (DP), normalized disparate impact [15] ($\overline{\text{DI}}$), statistical parity subgroup fairness [46] (SPSF), false positive subgroup fairness [46] (FPSF), equalized odds false positive [24] (EOFP), equalized odds true positive [35] (EOTP), and bias amplification [99, 108] (BA). For all bias metrics, 0 indicates the fairness condition is satisfied, i.e., a completely fair model, and 1 indicates a completely unfair model. Details of these metrics are in Appendix A.

---

[1]`https://github.com/lin-tan/fairness-variance/`

Table 1: Evaluated mitigation techniques and baselines

| Task | Technique | Debiasing Algorithm | Network | Tgt |
|------|-----------|---------------------|---------|-----|
| Multi-class classification in CIFAR-10S [99] (1) | S-Base | Baseline | ResNet-18 | BA |
| | S-RS | Data Rebalancing | | |
| | S-UC | Uniform confusion loss | | |
| | S-GR | Gradient reversal | | |
| | S-DD1 | Sum of probabilities | | |
| | S-DD2 | Max of prob. w/ prior shift | | |
| | S-DD3 | Sum of prob. w/ prior shift | | |
| | S-DD4 | RBA debias | | |
| | S-DI1 | Domain independent, conditional | | |
| | S-DI2 | Domain independent, sum | | |
| Multi-label classification in MS-COCO [98] (2) | C-Base | Baseline | ResNet-50 | DP (BA*) |
| | C-R1 | Data rebalancing, M/F = 1 | | |
| | C-R2 | Data rebalancing, M/F < 2 | | |
| | C-R3 | Data rebalancing, M/F < 3 | | |
| | C-A4 | Adv training at 4th conv block | | |
| | C-A5 | Adv training at 5th conv block | | |
| Multi-class classification in imSitu [98] (3) | I-Base | Baseline | ResNet-50 | DP (BA*) |
| | I-R1 | Data rebalancing, M/F = 1 | | |
| | I-R2 | Data rebalancing, M/F < 2 | | |
| | I-R3 | Data rebalancing, M/F < 3 | | |
| | I-A4 | Adv training at 4th conv block | | |
| | I-A5 | Adv training at 5th conv block | | |
| Binary classification in CelebA [59] (4) | A-Base | Baseline | ResNet-18 | EO |
| | A-L2 | L2 penalty | | |
| | A-ALM | FairALM | | |
| Binary classification in CelebA [47] (5) | N-Base | Baseline | NIFR | DP |
| | N-Flow | Conditional Flow | | |

**Per-class and overall bias:** Bias calculated on each class is called *per-class bias* and the bias for each model is the *overall bias*. For example, there are ten per-class biases in models trained on CIFAR-10 (one per class). The overall bias of the model is the average of all per-class biases.

**Multi-(task)label classification:** We treat each task label as an independent binary classification, and the overall bias value of the model is the average across all labels.

## 2.3 Reproduced studies

We conduct a literature survey of published papers from the recent five years (2016–2020) of the top ML (NeurIPS/NIPS, ICML, AAAI, and FAccT), and computer vision (CVPR, ICCV, and ECCV) conferences (Section 4). Out of the 32 papers relevant to the fairness of DL systems, only four [47, 59, 98, 99] have a working replication package so that we can reproduce their results. These papers study the fairness of DL systems for the five tasks described in Table 1 (expanded details in Appendix B.1). Overall, we replicate 27 different techniques from four state-of-the-art papers on five classification tasks. In the next section, we introduce each technique and detail our experiments.

## 2.4 Techniques

Table 1 lists the 27 techniques that we reproduce for our study. Column *Task* lists the tasks on which these techniques are applied. Column *Technique* lists the technique abbreviations we used. Column *Debiasing Algorithm* lists the debiasing algorithms used for each technique. Column *Network* lists the DL network used to train the models. Finally, column *Tgt* indicates which bias metric the corresponding debiasing algorithm is designed to reduce.

Five techniques (S-Base, C-Base, I-Base, A-Base, and N-Base) are baselines (i.e., do not use any debiasing algorithm), and the other 22 use debiasing algorithms. Techniques starting with "S" are from [99]; techniques starting with "C" and "I" are from [98] trained for tasks (2) and (3) respectively; techniques starting with "A" are from [59]; and techniques starting with "N" are from [47].

Apart from five baseline techniques, the remaining algorithms can be grouped into three categories including data rebalancing, fairness through blindness, and fairness through awareness. Details can be found in Appendix B.2.

**Optimization target for tasks (2) and (3)**: The study [98] uses "bias amplification" as their optimization goal. However, current literature does not use this term consistently. The original study definition (noted as BA*) is different from the one proposed by Zhao et al. [108] (noted as BA) and BA* is more analogous to the DP metric. Therefore, we denote this optimization target as *DP (BA*)*.

### 2.5  Variance experiments setup

Following the prior work [71], we use *FIT runs* to measure the variance of trained model fairness. For each technique, all the training runs are executed with the same training data (also the original training/test split), hyper-parameters, and optimizers. With the fixed seed, all training runs also have the same order of data and the same initial weights. We perform 16 FIT runs with the same random seed for each technique, and then evaluate the fairness of the trained models using seven bias metrics.

Since there is no theoretical bound on the variance caused by DL software implementations, and each set of 16 FIT runs can only be viewed as a sample of the real distribution, we use statistical tests, including Mann-Whitney U-test, Cohen's $d$ value, and Levene's test to ensure the statistical significance of our findings. Details of these statistical tests can be found in Appendix B.3.

## 3  Findings

We evaluate the fairness variance of 27 techniques (22 mitigation and five baseline) using seven bias metrics. These are 189 experiments (technique-metric pairs). To evaluate the impact of DL software implementation on different bias metrics, we perform 16 FIT runs for each of the 27 techniques, which produce 432 models. Each experiment generates one **bias value** per run (i.e., 16 bias values per experiment), which is a total of 3,024 bias values. The training time of the 189 experiments is 7,117 hours (**about 9.8 months**). Details of the hardware and software environment are in Appendix B.4.

To measure the difference across multiple FIT runs, we use two metrics: (1) the difference in percentage point as *MaxDiff* (maximum difference) and (2) the standard deviation in percentage point as *STDEV*. *MaxDiff* is the range of the bias values from a set of 16 FIT runs. For example, the *MaxDiff* of 16 runs, whose bias values are between 20.0%–30.0% inclusive, is 10.0%.

### 3.1  RQ1: How much variance of fairness does DL software implementation introduce?

Table 2 shows the fairness variance introduced by software implementation. Row *Technique* lists the techniques we focus on for this RQ. Row *Metric* shows the bias metrics used for evaluation. Row *Overall* shows the bias for the overall technique. Row *Per-class* shows the results for the class within each technique that has the most variance.

Table 2 shows that software implementation impacts all bias metrics. For example, we observed a maximum difference in DP with technique A-L2 of 12.6% (27.3%–39.9%), i.e., the most unfair model has 12.6% more gender bias measured by DP than the fairest model. Also, in technique S-GR, we observed a maximum difference in $\overline{\text{DI}}$ of 11.8% (21.3%–33.1%). Software implementation solely causes such variance, and thus cannot be eliminated by fixing the random seeds.

**The per-class variance of fairness and single class fairness failure** could be problematic if a class is privileged. For example, in a DL model for loan applications, the approval is the privileged class. A difference of 100% for $\overline{\text{DI}}$ means that in one run, the model is completely fair for all races (protected groups) to get approval. In another run, none of the applicants of a particular race gets approved, which is 100% bias against this minority group, even if the overall fairness variance is small.

Table 2: Maximum difference (MaxDiff) and standard deviation (STDEV) on the seven bias metrics for S-GR, A-L2, C-Base, and C-A5. All numbers are in percentage points.

| Metric | | DP | $\overline{\text{DI}}$ | SPSF | FPSF | EOFP | EOTP | BA | DP | $\overline{\text{DI}}$ | SPSF | FPSF | EOFP | EOTP | BA |
|---|---|---|---|---|---|---|---|---|---|---|---|---|---|---|---|
| **Technique** | | | | | **S-GR** | | | | | | | **A-L2** | | | |
| **Overall** (%) | **MaxDiff** | 1.7 | **11.8** | 0.9 | 0.5 | 1.2 | 7.2 | 4.2 | **12.6** | 8.0 | 6.0 | 1.6 | 7.2 | 7.2 | 11.6 |
| | **STDEV** | 0.5 | 3.5 | 0.3 | 0.2 | 0.4 | 2.0 | 1.3 | 2.9 | 2.4 | 1.4 | 0.5 | 2.1 | 2.1 | 3.1 |
| **Per-class** (%) | **MaxDiff** | 7.3 | 50.1 | 3.6 | 2.5 | 5.7 | 34.6 | 19.3 | 12.6 | 31.1 | 6.0 | 3.5 | 14.3 | 14.3 | 15.9 |
| | **STDEV** | 1.9 | 12.7 | 1.0 | 0.7 | 1.4 | 8.1 | 5.0 | 2.9 | 8.8 | 1.4 | 1.2 | 4.8 | 4.8 | 4.2 |
| **Technique** | | | | | **C-Base** | | | | | | | **C-A5** | | | |
| **Overall** (%) | **MaxDiff** | <0.1 | 2.6 | <0.1 | <0.1 | <0.1 | 1.9 | 1.4 | 0.1 | 2.4 | <0.1 | <0.1 | <0.1 | 2.6 | 3.1 |
| | **STDEV** | <0.1 | 0.8 | <0.1 | <0.1 | <0.1 | 0.6 | 0.4 | <0.1 | 0.6 | <0.1 | <0.1 | <0.1 | 0.6 | 0.8 |
| **Per-class** (%) | **MaxDiff** | 1.6 | **100.0** | 0.8 | 0.5 | 1.1 | 33.3 | 33.3 | 3.7 | **100.0** | 1.8 | 1.0 | 2.1 | 43.9 | 65.4 |
| | **STDEV** | 0.5 | 38.9 | 0.3 | 0.2 | 0.4 | 11.7 | 8.1 | 0.9 | 28.2 | 0.4 | 0.2 | 0.5 | 10.8 | 21.8 |

---

**Finding 1**: Across 16 FIT runs, software implementation causes a maximum difference of up to 12.6% in the DP bias metric, and up to 11.8% in the $\overline{\text{DI}}$ metric. A complete single-class fairness failure is possible, where one run produces a model with 0% bias while another with 100% bias.

---

Additional results of this RQ can be found in Appendix D.1. We discuss the implications of these results for researchers and practitioners in Section 5.

## 3.2 RQ2: How does software implementation affect debiasing algorithms?

To quantify the impact of *software implementation* on debiasing algorithms, we use the Mann-Whitney U-test and Levene's test to check whether the *average bias value* and the *variance of bias value* after debiasing are statistically different from those of its baseline respectively. We normalize each technique's bias values before Levene's test to eliminate the impact of the average value [5, 28, 77, 91].

For 22.1% (34 of 154) of the experiments (the comparison of 22 mitigation techniques with their baseline techniques evaluated using seven bias metrics), the mitigation techniques' bias values overlap with those of the baselines. For example, the range of EOTP bias values for the A-ALM debiasing is 19.6%–24.3%, while its baseline's (A-Base) range is 22.6%–24.7%.

For 9.1% (14 of 154) of the experiments, the mitigation technique's average bias metric is higher than the baseline, yet a single run's bias value is lower than a single run of the baseline. Thus, using a single run, one may incorrectly conclude that the mitigation technique is fairer, but on average, the mitigation technique is less fair. For example, *S-Base* (7.4% BA) is fairer than *S-GR* (8.7% BA) on average of 16 FIT runs ($p = 0.0011$ in U-test). However, one may obtain a fairer model with S-GR from one training run (e.g., BA of 6.1% for S-GR vs. 7.8% for S-Base). Therefore, DL fairness researchers should perform multiple runs and check statistical tests to ensure the effectiveness and validity of the debiasing algorithms. Our Levene's test results also confirm that the variance of bias values of *S-GR* is statistically higher than that of *S-Base* ($p = 0.0005$).

---

**Finding 2**: The effectiveness of the debiasing algorithms can be inaccurate with only one fixed-seed training run.

---

Since there is no consensus on which bias metric to use [19, 49], and some bias constraints cannot be satisfied simultaneously [50], we study if and how a debiasing algorithm reduces biases measured by one bias metric but increases biases measured by a different metric.

For the 154 experiments above, we conduct a series of Mann-Whitney U-tests on the bias values obtained. For each test that exhibits a statistically significant difference, we compute effect size using Cohen's $d$ value between the two sets of bias values (baseline and mitigation) involved. The $d$ values tell us whether each mitigation technique has a meaningful effect or not. Raw values before and after mitigation are available in the supplementary material.

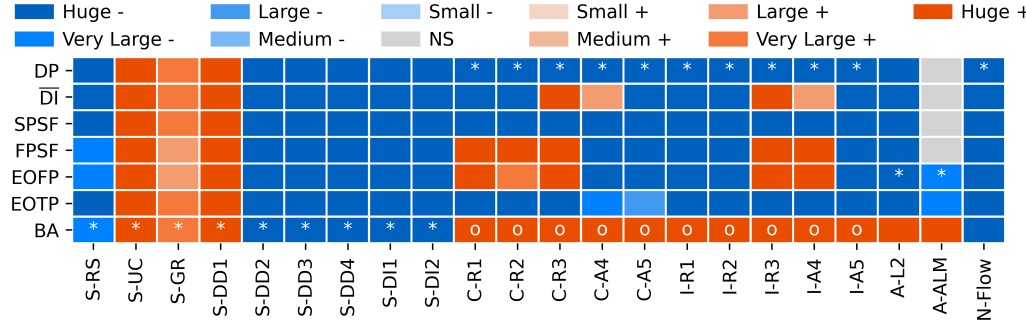

Figure 1: Effect size of *bias changes* between before and after mitigation application. Blue is a bias decrease (good), orange an increase (bad), and grey not statistically significant (NS).

Figure 1 shows the effect sizes of bias value between 22 mitigation techniques and their corresponding baselines. The x-axis and y-axis labels are the abbreviations for the techniques and bias metrics. If a mitigation technique statistically reduces the bias value on a bias metric (with respect to its baseline), the corresponding block is blue. Oppositely, the block is orange. The block is grey for mitigation techniques that do not induce a statistically significant difference from its baseline. The different shades of color show the standard grouping and interpretation of the effect size, such as "Huge" and "Very large". Figure 1's legend shows the color details. Blocks with a "*" are the bias metric that the mitigation technique is designed to reduce in the original research (e.g., BA for S-DI2 and DP for N-Flow). An "o" is a reminder that the optimization target of the mitigation techniques that begin with "C" and "I" is better captured by the DP metric (BA* in the original research, see Section 2.4).

Figure 1 shows that most (96 out of 103) mitigation techniques that induce a statistically significant improvement on the bias value have a huge effect (deep blue block). However, the vast majority (68.2%, 15 of 22) of mitigation techniques fail to reduce biases measured by all seven bias metrics. For example, A-L2 and A-ALM aim to improve bias metric EOFP. Although they succeeded, they also increase biases for BA.

This experiment shows that for most of the mitigation techniques, bias metrics may not be compatible with each other. Practitioners applying mitigation techniques must understand the techniques' optimization goal and use statistical tests.

> **Finding 3**: Most of (15 out of 22) the mitigation techniques tested increase at least one of the bias metrics: all 15 of them fail to improve the newly introduced bias amplification metric.

### 3.3 RQ3: How does software implementation affect the cost of mitigation techniques?

The most known cost of mitigation techniques is the trade-off between fairness and model accuracy [25, 100]. However, with the software implementation taken into consideration, mitigation techniques could introduce new costs including more variance of model accuracy and fairness.

There are three dimensions to the cost of mitigation techniques: lower accuracy, higher accuracy variance, and higher fairness variance. We excluded the techniques that increase bias for all metrics compared to its baseline. For accuracy, we use U-test to check if the reduction is statistically significant. For the variance of both model accuracy and model bias, we use Levene's test.

Figure 2 shows the cost of mitigation techniques on fairness variance (Levene's test). Labels on x-axis and y-axis are the same as those in Figure 1. Blue and orange blocks respectively indicate lower and higher variance than the baseline. Grey blocks mean no statistically significant change in the variance of bias values, and "X" indicates mitigation technique fails to improve the bias metric tested. For example, the orange [N-Flow, DP] block shows that N-Flow generates models whose variance of DP bias values is higher than that of the baseline, and the comparison results are statistically significant.

To study the relationship among the three types of cost, we further examine the 154 experiments on mitigation techniques. Most of the experiments (103) reduce biases. The majority of them (70 out of 103) achieve bias reduction with a cost in one or more of the three dimensions. Eight experiments come with all three costs: lower accuracy, higher accuracy variance, and higher fairness variance, and are from A-L2 and A-ALM. However, there is no correlation between accuracy variance and bias

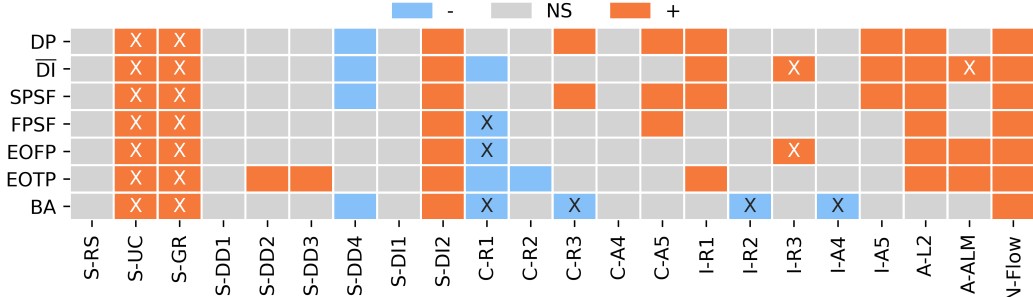

Figure 2: Cost of bias mitigation techniques on the *variance* of bias metrics after normalization (Levene's tests). Blue denotes a decrease in variance (good), orange an increase in variance (bad), and grey not statistically significant (NS).

variance. Interestingly, mitigation technique S-DD4 (RBA [108] debiasing) successfully reduces all of the seven bias metrics without sacrificing model accuracy or introducing more variance.

DL researchers and practitioners might overlook the cost of fairness variance when optimizing a model toward a particular fairness goal if they are unaware of the variance caused by DL software [71].

> **Finding 4**: Out of 154 experiments with debiasing algorithms, 103 experiments show the applied algorithms reduce the bias values. In a majority (68.0%, or 70) of these 103 experiments, the debiasing algorithms have a negative impact on the model (36 increase the variance of bias value, 54 decrease the accuracy, and 15 increase the variance of accuracy, not exclusively).

# 4   Literature survey

We conduct a literature survey from the top ML conferences to evaluate the awareness of fairness variance of DL systems.

**Paper selection criteria:** We extract papers from the last five years (2016-2020) of the top ML (NeurIPS/NIPS, ICML, and AAAI), and computer vision (CVPR, ICCV, and ECCV) conferences. We also include FAccT and its predecessor FAT/ML workshop, an interdisciplinary venue of ML and fairness.

We only consider papers that contain fairness-related keywords (fair, bias, discriminate, aware, balance, disparity, opportunity, audit, gender, stereotype, race, amplification, ethic) in their titles using the Snowball Stemmer [73]. Then, we read the abstract and the paper and keep relevant papers involving the fairness of a classification task. Out of 24,186 papers, we extracted 274 papers studying fairness, including 32 dedicated to DL systems' fairness.

**Paper survey results:** While still taking only a small portion of accepted papers, the number of papers related to fairness has been increasing over the last five years. Researchers need to be aware of DL models' fairness variance caused by DL software implementations.

> **Finding 5**: DL fairness is gaining more attention in the research community since 2017. The number of papers related to DL fairness in top ML and computer vision conferences has increased 750% since 2017 (two papers in 2017, 17 papers in 2020).

To understand whether researchers are aware of the fairness variance of DL training, we manually check whether the papers on the fairness of DL systems use multiple identical runs in their evaluation.

> **Finding 6**: Of 32 papers relevant to the fairness of DL systems, only 34.4% (11 out of 32) of them use multiple identical runs to evaluate their proposed approaches.

A complete list of 32 papers surveyed in this section can be found in Appendix D.3.

# 5 Implications and suggestions

**Research validity:** When comparing the results of a proposed method (e.g., a new debiasing algorithm or a new DL network), it is necessary to run statistical tests to confirm any improvements.

**Testing for fairness in online learning:** When testing an online-learning software system for fairness (where models are continuously updated), such as identity-inference-based advertisement targeting systems [7, 70, 101], multiple identical runs are necessary, and statistical test results should be reported to measure the variance of bias values caused by the DL software implementation.

**Research reproducibility:** Given the increasing number of papers on fairness in AI [6, 32, 48, 49, 63, 65, 79, 105], software engineering [10, 14, 16, 94], and other conferences [2, 18, 52], reproducibility becomes increasingly important [4, 78]. Following prior efforts [43, 56, 72] that improve the reproducibility of DL methods, researchers may want to develop such guidance for fairness, e.g., considering multiple bias metrics, as variance levels are different for different bias metrics. Proper documentation of the software and hardware development environment and configuration parameters would also facilitate the reproduction of research results. As shown in Section 3.1, software implementation alone causes software fairness to vary by up to 12.6%, which is much higher than the 2.9% accuracy variance discovered by previous work [71].

**Balancing hidden costs of debiasing:** Since variance is a hidden cost for debiasing algorithms and could harm the fairness performance of online learning algorithms (Section 3.3), there is usually a trade-off between model accuracy, fairness, and variance. It is important that DL researchers and practitioners understand debiasing algorithms' costs, especially variance, and strike a balance among them.

**Deterministic DL software for model tuning and debugging:** Building deterministic DL libraries could be a research topic to improve model tuning and the debugging of DL systems. For a partial solution, one could use the deterministic mode (Appendix E.1) when it is provided by DL libraries.

**Maximizing fairness through multiple runs:** To take advantage of software variance, practitioners could train multiple times to select the fairest model. In this case, proper testing of the DL model is recommended to ensure the satisfaction of regulatory requirements, considering the pre-deployment and real-world data gap.

**Selection of bias metrics:** In most cases, a debiasing algorithm does not reduce bias on all targets (Section 3.2), and multiple fairness constraints could contradict each other [49, 50]. Therefore, developers and testers for DL systems should carefully select fairness requirements. Furthermore, optimizing on one bias metric may not be enough as it might make the model less fair on other metrics. DL researchers should consider the task and use the appropriate bias metric. In the strictest sense, one may need to use multiple bias metrics to ensure fairness across multiple dimensions.

# 6 Limitations and discussions

**Limitation**: It is known that gender bias exists in language embeddings [108]. We only study image classification tasks because they are a better fit for our scope of group fairness and bias metrics on image classification tasks are better defined. Also, we only empirically obtain the upper bound of fairness variance and did not include any theoretical proof. We leave both limitations as future work.

All 27 techniques reproduced are implemented in PyTorch instead of other popular DL frameworks such as TensorFlow and MXNet. We decide to reuse the implementation from the original authors as it is error-prone to reimplement all the techniques in another framework. Since modern DL frameworks share many implementation concepts (e.g., parallelism and auto-tuning, see Appendix E.1) as well as underlying libraries such as cuDNN and CUDA, they are likely to suffer from the same variance problem.

**Deterministic mode of DL frameworks**: A deterministic mode has been introduced to modern DL frameworks such as PyTorch and TensorFlow to tackle the nondeterminism from software implementations. However, at the time of writing, the deterministic mode cannot eliminate the fairness variance described in this paper (See Appendix E.2 for details). In addition, as the survey in our prior work [71] shows, over 80% of the respondents are unaware or unsure about the nondeterminism

caused by software implementations. DL researchers and practitioners would not use the deterministic mode if they are unaware of the fairness variance caused by DL software.

**Negative social impact**: With FIT runs to estimate the fairness variance of neural networks, our proposed approach could increase the power consumption and $CO_2$ emissions of DL model training. However, such additional evaluation is required for experimental validity and fairness compliance. In other words, the negative social impact of not performing multiple training runs includes disproportionally hurting certain races or genders. Our paper raises awareness of this issue and calls for new solutions to strike a balance. We are working on theoretically quantifying the upper bound of fairness variance and cheaper ways to estimate/bound variance.

# 7 Related work

**Anecdotal evidence in fairness variance:** Agrawal et al. [2] apply debiasing on the random forest classifiers and test for fairness variance with different random seeds and data splits for cross-validation. They find that debiasing algorithms could increase the fairness variance introduced by algorithmic nondeterminism. Different from their work, we focus on DL models and variance caused by DL software implementations, which cannot be removed by fixing random seeds.

Friedler et al. [30] investigate the stability of debiasing algorithms by measuring the standard deviation of fairness metrics over different dataset splits. However, they overlook the nondeterminism from the DL software implementations and only perform one training run for each split.

**Fairness studies of ML systems:** The fairness of ML and DL systems have gained attention and several studies pointed out fairness issues in datasets (e.g., CelebA [81], GitHub [41], and StackOverflow [61]) and major ML systems such as language models [13, 92, 95], face recognition [89], and sentiment analysis [88]. Such studies demonstrate that ML systems can be unfair and highlight the importance of fairness. Yet to the best of our knowledge, we are the first to study the impact of DL software implementation on fairness.

**Bias metrics:** Many metrics [6, 15, 19, 24, 26, 31, 35, 44, 46, 67, 82, 102, 103] have been proposed to measure biases of ML systems. Most of them can be applied to DL systems too. We evaluate fairness using seven of the most popular metrics and show that they are all impacted by DL implementation variance. None of the existing metrics considers such variance in their definition.

**Debiasing algorithms:** To mitigate biases of DL systems, many debiasing algorithms or bias resistant networks have been proposed and evaluated [10, 15, 16, 17, 45, 47, 54, 59, 66, 83, 98, 99, 106]. These techniques are important to improve the fairness of DL systems but their evaluation can be affected by fairness variance introduced by DL software implementations. Our study shows that the impact of some debiasing algorithms is not consistent during different identical runs.

# 8 Conclusion

This work studies the fairness variance of DL systems caused by software implementations. We examine five tasks on three networks and four datasets using 22 mitigation techniques. We find up to 12.6% difference in bias values across FIT runs and 68.0% of the experiments reduce model biases with cost in fairness variance, accuracy, or accuracy variance. Our paper survey shows that only 34.4% of the papers use multiple identical runs to evaluate the proposed approaches. We call for the awareness of the fairness variance of DL systems caused by software implementations.

## Acknowledgments

The authors thank the anonymous reviewers for their constructive comments as well as the area chair and the senior area chair for overseeing the review process. This work has been partially supported by NSF 2006688 and a J.P. Morgan AI Faculty Research Award. We also thank NSERC for funding support. Any opinions, findings, and conclusions in this paper are those of the authors only and do not necessarily reflect the views of the sponsors.

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
