# A Bias Metrics

## A.1 Symbols

For a binary classification task, we abbreviate the content in the confusion matrix as $TP$, $FN$, $TN$, and $FP$. Positive ($P$) instances include both $TP$ and $FN$, while negative ($N$) instances include both $TN$ and $FP$.

*Prediction positive rate* ($PPR$), *true positive rate* ($TPR$), and *false positive rate* ($FPR$) are derived from the confusion matrix and describe a binary classifier. $PPR$ is the ratio of instances classified as positive to the instances in the whole dataset ($\frac{TP+FP}{P+N}$). $TPR$ is the ratio of $TP$ to the positive instances ($\frac{TP}{P}$). $FPR$ is the ratio of $FP$ to the negative instances ($\frac{FP}{N}$).

Each protected group $g$ takes $\Pr[g]$ ($\frac{P_g+N_g}{P+N}$) of the dataset. We note the confusion matrix and the derived metrics for group $g$ by adding an annotation to the terms (e.g., $TP_g$, $FP_g$, $P_g$, $N_g$, $PPR_g$, $TPR_g$, and $FPR_g$).

## A.2 Selected Bias Metrics

**Condition 1**: Statistical Parity (SP) [26]: $PPR_0 = PPR_1$

- Demographic Parity [15]: **DP** $= |PPR_0 - PPR_1|$
- Normalized Disparate Impact [15]: $\overline{\textbf{DI}} = 1 - \min(\frac{PPR_0}{PPR_1}, \frac{PPR_1}{PPR_0})$
- Statistical Parity Subgroup Fairness [46]: **SPSF** $= \sum_{g \in G} \Pr[g] \times |PPR - PPR_g|$

**Condition 2**: Predictive Equality (PE) [19]: $FPR_0 = FPR_1$

- False Positive Subgroup Fairness [46]: **FPSF** $= \sum_{g \in G} \Pr[g] \times |FPR - FPR_g|$
- Equalized Odds (False Positive) [24]: **EOFP** $= |FPR_0 - FPR_1|$

**Condition 3**: Equality of Opportunity (EOp) [35]: $TPR_0 = TPR_1$

- Equalized Odds (True Positive) [35]: **EOTP** $= |TPR_0 - TPR_1|$

The last bias metric, bias amplification [99, 108] does not fit any existing fairness conditions.

- **BA** $= \left| \frac{TP_{\bar{g}} + FP_{\bar{g}}}{TP + FP} - \frac{P_{\bar{g}}}{P} \right|$, where $\bar{g} = \underset{g}{\arg\max} \frac{P_g}{P}$

# B Details of Reproduced Studies

## B.1 Tasks

**(1) Multi-class image classification in CIFAR-10S [99]:** This study classifies images from the CIFAR-10S dataset into ten task labels. CIFAR-10S is designed to study fairness issues and debiasing algorithms. Its protected label is whether the images are color or grayscale. Nine different debiasing algorithms (and a baseline) have been evaluated with this dataset using the popular ResNet-18 network [36].

**License**: Either CIFAR-10 [51] or CIFAR-10S [99] datasets did not specify the type of license used.

**(2) Multi-label classification in MS-COCO [98]:** The MS-COCO dataset [55] contains images with several objects (i.e., several labels in each image). In their fairness study, Wang et al. [98] find that this dataset can leak gender information (i.e., gender is the protected label) and study five different debiasing algorithms (plus one baseline) on ResNet-50 [36] models trained on MS-COCO.

**License**: The annotations [20] released under Creative Commons Attribution 4.0 License, and the images from MS-COCO datasets must abide by Flicker Terms of Use [93].

**(3) Multi-class classification in imSitu [98]:** This task is the second task used for benchmarking debiasing algorithms in previous work [98]. The task labels in imSitu [104] are actions (e.g., cooking), and the protected label is gender. Wang et al. [98] compare the same five debiasing algorithms as for task (2) with the same network.

**License**: The images of the imSitu dataset is gathered from Google Image Search [104], and the annotations are labeled by Amazon Mechanical Turk [104]. The authors did not specify the type of license used.

**(4) Binary classification in CelebA [59]:** Lokhande et al. [59] propose a new debiasing algorithm (FairALM) and compare it to a baseline and another debiasing algorithm on a ResNet-18 model trained on the CelebA dataset [57]. CelebA contains faces of celebrities with several binary task labels and two protected labels (gender and youth). Task (4) uses the task label "attractive" and the protected label "gender".

**License**: The CelebA dataset is available for non-commercial research purposes only [57].

**(5) Binary classification in CelebA with NIFR network [47]:** For the final task, we reproduced another work [47] that uses the CelebA dataset. Key differences with task (4) are the use of NIFR network, different debiasing algorithms, and a different task label ("smile" instead of "attractive").

## B.2   Debiasing Algorithms

**Data rebalancing** rebalances the training data to either be perfectly balanced (techniques S-RS, C-R1, I-R1) or within a certain ratio between the number of instances from each protected group (techniques C-R2, C-R3, I-R2, I-R3).

**Fairness through blindness** removes the protected label and creates an intermediate representation for each instance. Then a downstream classifier is trained to predict the task label on this representation. Adversarial training is the most popular debiasing algorithm within such category (techniques S-UC, S-GR, C-A4, C-A5, I-A4, I-A5). Recently, invertible neural networks [42] are also used to remove sensitive information (technique N-Flow).

**Fairness through awareness** consists of two categories: (1) the debiasing algorithm uses a loss function on protected labels to penalize the classifier's biases (techniques A-L2, A-ALM); and (2) the debiasing algorithm involves a fine-grained classifier that predicts the task label and the protected label of each instance. Before the final prediction on the task label, the predicted protected label is used to remove possible biases in the final prediction (techniques S-DD1, S-DD2, S-DD3, S-DD4, S-DI1, S-DI2). One concern of this debiasing algorithm is the waste of classification power to discriminate the boundaries between each protected group [99].

## B.3   Statistical Tests

**Mann-Whitney U-test:** One-sided U-test is used in this paper to compare the mean of two sample distributions. U-test is chosen instead of T-test because it does not require normality in the sample distribution. For two experiments $A$ and $B$, the null hypothesis is that the mean bias value of 16 runs in experiment $A$ is similar to that in experiment $B$. We choose a significance level of 5%, i.e., if we get a p-value < 0.05, we are 95% confident that experiment $A$ has a statistically different mean bias value than experiment $B$.

**Cohen's $d$:** In addition to the p-value from U-test, we use Cohen's $d$ to quantify the impact of debiasing algorithms. We use the empirical interpretation of effect size [85] to represent Cohen's $d$ values (e.g., Cohen's $d$ of 0.8 represents a "large" impact while a 2.0 represents a "huge" impact).

**Levene's test:** We use Levene's test to compare the magnitude of the variance between experiment runs. For two experiments $A$ and $B$, the null hypothesis is that the variance of the bias value from 16 runs of experiment $A$ is similar to that of experiment $B$. The significance level is also 5%.

## B.4   Hardware and Software

We conduct our experiments on a server with two Xeon Gold 5120 CPUs (56 cores in total) and 384 GB of memory. We use an RTX 2080Ti graphics card with 11GB of memory for each training run.

Table 3: Confusion matrix and bias metric values for a biased image classifier. "Sport" and "Cook" are the number of instances of real labels of "Sport" and "Cook" activity in the image. "Pred-Sport" and "Pred-Cook" are the number of instances of the predicted labels. "Male" and "Female" are the protected groups.

| | Female (200) | | Male (200) | | Total |
|---|---|---|---|---|---|
| | Pred-Sport | Pred-Cook | Pred-Sport | Pred-Cook | |
| Sport | 80 | 20 | 60 | 40 | 200 |
| Cook | 10 | 90 | 70 | 30 | 200 |
| **Total** | 90 | 110 | 130 | 70 | 400 |

| **Metric** | DP | $\overline{\text{DI}}$ | SPSF | FPSF | EOFP | EOTP | BA |
|---|---|---|---|---|---|---|---|
| **Value (%)** | 20.0 | 33.6 | 10.0 | 20.0 | 40.0 | 40.0 | 10.1 |

We use CUDA 10.0 and cuDNN 7.6, TensorFlow 1.15, and PyTorch 1.7 DL libraries. Each run is isolated using Docker.

## C  Bias Metrics Running Example

We use the running example in Table 3 to explain SPSF, DP, and BA in detail. Table 3 shows the prediction results from a biased binary classifier and its bias values using the seven metrics. The classifier assigns labels "Sport" and "Cook" with respect to the activity in the image, which is a common classification task with DL [37].   For example, for the 200 images containing a woman, 80 images doing sport are classified as "Sport", while another 20 of them are misclassified as "Cook". The protected label is gender, i.e., whether the person inside the image is "Male" or "Female" (nonbinary genders exist but are not considered in this example). The ground-truth labels of the dataset are balanced, i.e., 100 instances for each task label in each protected group.   Bias metrics are calculated on both task labels, and the average of the two is the bias of the model.

We choose to balance the protected group in the sample dataset because we would like to stress that even with a balanced dataset, the trained model could still be biased. With unbalanced groups, the bias could be even bigger [99, 108], and the biases from the best scenario are already worrisome.

**(1) Demographic Parity (DP) [15]**: DP is measured by the difference of the $PPR$ between different protected classes. Without losing generality, we consider "Sport" the positive class in the binary classifier. Following the DP formula in Appendix A.2, for the "Sport" class, the $PPR_{\text{female}}$ is 45.0% (90 / 200), and $PPR_{\text{male}}$ is 65.0% (130 / 200), a difference of 20.0%, i.e., the male group is 20.0% more likely to be labeled as "Sport". The DP bias metric for the "Sport" class is 20.0% ($110/200 - 70/200$). Similarly, the DP for the "Cook" class is 20.0%. Therefore, the model has a DP bias of 20.0%, which is the average of the two per-class biases.

**(2) Disparate Impact (DI, $\overline{\text{DI}}$) [15]**: DI measures the ratio of $PPR$ between protected groups. The less protected groups are discriminated against, the closer DI gets to 1. In this paper, we measure Normalized DI ($\overline{\text{DI}}$) as the difference between 1 and DI ($\overline{\text{DI}} = 1 - \text{DI}$) to be consistent with other bias metrics (i.e., the model is completely fair when $\overline{\text{DI}} = 0$, and unfair when $\overline{\text{DI}} = 1$).

**(3) Statistical Parity Subgroup Fairness (SPSF) [46]**: SPSF is also known as the equality of classification rates. The goal of SPSF is that all protected classes have the same $PPR$ as the whole dataset.

$PPR$ is 55.0% (220 / 400) for the dataset.  $PPR_{\text{male}}$ is 65.0% (130 / 200), 10.0% more likely to be predicted as  positive. $PPR_{\text{female}}$ is 45.0% (90 / 200), 10.0% less likely to be predicted positive. The difference in $PPR$ for each group is both 10.0%, and each group makes up 50.0% of the instances (Pr[male] and Pr[female]). Therefore, the SPSF bias for the "Sport" class is 10.0%. Similarly, the SPSF bias for the "Cook" class is also 10.0%. The overall SPSF bias is the weighted average of the two: 10.0%.

**(3) False Positive Subgroup Fairness**: (FPSF) [46] Similar to SPSF, FPSF requires each protected class to have the same false positive rate as the whole dataset. Compared with SPSF, this metric considers the accuracy of the classifier. It should be used with SPSF together to measure the fairness of the classifier.

**(5) and (6) Equalized Odds (EO) [24, 35]**: Equalized Odds was originally introduced in recidivism prediction. There are two variants of this metric. One requires the same $TPR$, and the other requires the same $FPR$ across all protected classes. We name these two variants as Equalized Odds - True Positive (EOTP) and Equalized Odds - False Positive (EOFP) by their definition.

The female group has a $TPR$ of 80% (80 / (80 + 20)), and the male group has a $TPR$ of 60% (60 / (60 + 40)). The difference between $TPR_{\text{female}}$ and $TPR_{\text{male}}$ is 20%, which is a EOTP of 0.2. Similarly, the EOTP for "Cook" class is 0.6. And the classifier has a EOTP of 0.4.

Similar to the process in EOTP, the classifier has a EOFP of 0.4 on this dataset.

**(7) Bias Amplification (BA) [99, 108]**: Unlike the six bias metrics above, which only look at the outcome of a classifier, bias amplification separates the bias in the dataset and the bias caused by the algorithm. It assumes a pre-existing skew inside the dataset (e.g., gender bias), and measures how much the algorithm amplifies such skew.

The "Sport" class has 100 male ($\frac{P_{\text{male}}}{P} = 50.0\%$) and 100 female ($\frac{P_{\text{female}}}{P} = 50.0\%$) instances. In the prediction result, there are 90 female ($\frac{TP_{\text{female}}+FP_{\text{female}}}{TP+FP} = \frac{80+10}{140+80} = 40.9\%$) and 130 male ($\frac{TP_{\text{male}}+FP_{\text{male}}}{TP+FP} = \frac{60+70}{140+80} = 59.1\%$) instances. The prediction result for the "Sport" class is 9.1% more biased towards the male group than the dataset, and the model *amplifies bias* towards the male group by 9.1%, which is a BA of 9.1% for the "Sport" class.

Similarly, for the "Cook" class, the model *amplifies bias* toward the female group by 11.1% (BA of 11.1%). Therefore, the classifier has a BA of 10.1% (average of 9.1% and 11.1%), i.e., *amplifies bias* by 10.1%.

# D    Additional Results

## D.1    RQ1

Table 4 shows the aggregated results for all of the 27 techniques evaluated with $\overline{\text{DI}}$ metric. This table, together with Table 2, shows that fairness variance is not tied to a particular debiasing algorithm nor a particular bias metric. A per-class absolute maximum difference of 100% in $\overline{\text{DI}}$ metric occurred for multiple techniques (C-Base, C-R3, C-A4, and C-A5).

## D.2    RQ3

Figure 3 shows the cost of mitigation techniques on the model accuracy and the variance of accuracy. Mann-Whitney U-test confirms the cost in model accuracy and Levene's test confirms the cost in accuracy variance. The upper subgraph is the cost on model accuracy, and the lower subgraph is the cost on variance of model accuracy. A blue block indicates an increase in model accuracy or a decrease in the variance of model accuracy. Oppositely, an orange block is used. A grey block indicates no statistically significant change. An "X" inside the block indicates that the mitigation technique fails to improve any of the bias metrics tested.

## D.3    Literature Survey

Table 5 shows the detailed results of the 32 papers related to DL fairness in our literature survey. Column "Paper" cites the each paper, and column "Use multiple runs" indicates whether one paper uses multiple training runs to ensure validity or not. The other three columns indicate whether a paper reports the average value ("Mean"), standard deviation ("STDEV"), and range ("Range"). A "Y" means "yes", a "N" means "no", and a "-" indicates the column is not applicable for the paper.

From the table, only 11 out of 32 papers use multiple training runs to evaluate the proposed approaches. All of the papers that use multiple runs report mean values but some of them only report the standard deviation instead of range. We believe it is important to report all three statistical values as the range

Table 4: Maximum differences of $\overline{\text{DI}}$ values for all techniques

| Technique | Overall (%) | | Per-class (%) | |
|---|---|---|---|---|
| | MaxDiff | STDEV | MaxDiff | STDEV |
| S-Base | 2.0 | 0.5 | 7.0 | 1.9 |
| S-RS | 2.0 | 0.5 | 5.8 | 1.2 |
| S-UC | 5.1 | 1.7 | 18.4 | 4.6 |
| S-GR | **11.8** | 3.5 | 50.1 | 12.7 |
| S-DD1 | 2.3 | 0.6 | 5.4 | 1.5 |
| S-DD2 | 1.5 | 0.4 | 5.3 | 1.7 |
| S-DD3 | 1.4 | 0.4 | 5.7 | 1.7 |
| S-DD4 | 1.0 | 0.2 | 8.0 | 1.8 |
| S-DI1 | 1.8 | 0.5 | 5.0 | 1.4 |
| S-DI2 | 1.1 | 0.3 | 4.2 | 1.0 |
| C-Base | 2.6 | 0.8 | **100.0** | 38.9 |
| C-R1 | 1.0 | 0.3 | 50.0 | 19.5 |
| C-R2 | 2.8 | 0.8 | 62.5 | 21.1 |
| C-R3 | 3.7 | 1.1 | **100.0** | 29.9 |
| C-A4 | 3.7 | 0.8 | **100.0** | 22.8 |
| C-A5 | 2.4 | 0.6 | **100.0** | 28.2 |
| I-Base | 0.8 | 0.2 | 58.9 | 22.6 |
| I-R1 | 0.9 | 0.3 | 50.0 | 14.8 |
| I-R2 | 0.7 | 0.2 | 30.0 | 8.1 |
| I-R3 | 1.2 | 0.4 | 44.7 | 13.4 |
| I-A4 | 0.8 | 0.2 | 40.0 | 14.5 |
| I-A5 | 1.0 | 0.3 | 50.0 | 23.2 |
| A-Base | 1.7 | 0.5 | 5.8 | 1.8 |
| A-L2 | 8.0 | 2.4 | 31.1 | 8.8 |
| A-ALM | 4.5 | 1.2 | 9.3 | 3.0 |
| N-Base | 0.0 | 0.0 | 0.0 | 0.0 |
| N-Flow | 2.5 | 0.6 | 3.6 | 0.9 |

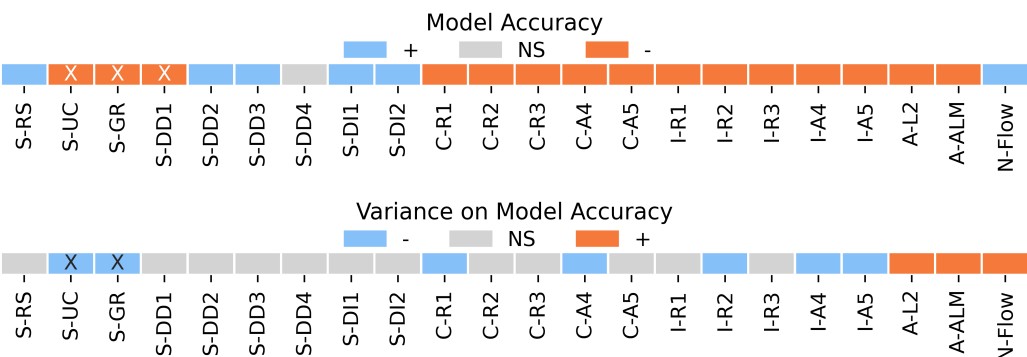

Figure 3: Cost of bias mitigation techniques on model accuracy and accuracy variance

could indicate the worst (or the best) case scenario, which is critical to the compliance requirements of the trained models.

# E   Discussion of DL Software Nondeterministic

## E.1   Causes

As our prior work [71] pointed out, nondeterminism in DL software can be divided into two parts: algorithmic nondeterminism and implementation-level nondeterminism.

Table 5: Result of literature survey. "Y" and "N" under each column indicate whether the corresponding paper uses multiple training runs or reports statistical values.

| Paper | Use multiple runs | Mean | STDEV | Range |
|---|---|---|---|---|
| Li et al. [54] | N | - | - | - |
| Min et al. [66] | N | - | - | - |
| Wang and Deng [97] | N | - | - | - |
| Wang et al. [99] | Y | Y | Y | Y |
| Quadrianto et al. [75] | Y | Y | Y | N |
| Wang et al. [98] | N | - | - | - |
| Lokhande et al. [59] | Y | Y | Y | Y |
| Sarhan et al. [83] | N | - | - | - |
| Rockwell and Fouhey [80] | N | - | - | - |
| Kehrenberg et al. [47] | N | - | - | - |
| Hendricks et al. [37] | N | - | - | - |
| Savani et al. [84] | Y | Y | N | Y |
| Vig et al. [95] | N | - | - | - |
| Nam et al. [69] | Y | Y | N | Y |
| Tan and Celis [92] | N | - | - | - |
| Grover et al. [33] | Y | Y | Y | Y |
| McDuff et al. [62] | Y | Y | N | Y |
| Creager et al. [21] | N | - | - | - |
| Pryzant et al. [74] | N | - | - | - |
| Guo et al. [34] | N | - | - | - |
| Dev et al. [22] | Y | Y | N | Y |
| Singh et al. [89] | N | - | - | - |
| Sen and Ganguly [87] | N | - | - | - |
| Adel et al. [1] | Y | Y | N | N |
| Nabi and Shpitser [68] | N | - | - | - |
| Black et al. [11] | Y | Y | Y | N |
| Ryu et al. [81] | N | - | - | - |
| Shen et al. [88] | N | - | - | - |
| Wadsworth et al. [96] | N | - | - | - |
| Hendricks et al. [38] | N | - | - | - |
| Blodgett and O'Connor [12] | N | - | - | - |
| Beutel et al. [8] | Y | Y | N | Y |

Algorithmic nondeterminism includes nondeterministic DL layers such as dropout, weight initialization, data augmentation such as random cropping and rotation, and batch ordering [71].

With fixed-seed training, we can completely eliminate algorithmic nondeterminism and have exactly the same training data, dropout behavior, and initial weight.

Implementation-level nondeterminism has two causes [71]:

1. DL software selects primitive operations at runtime. For example, cuBLAS automatically selects computation kernels with respect to buffer availability, and DL frameworks such as PyTorch choose the fastest CUDA kernel based on benchmark results before every training run.

2. Float-point calculations are not associative. For faster training and pre-processing, data is handled by the DL system in parallel processes, and such processes finish the local task at a different speed, resulting in different orders of data fed into the network. Neural network training on GPUs is also highly parallelized. Thread scheduling results in nondeterministic floating-point reduction orders, causing different results across multiple identical training runs.

### E.2 Deterministic Mode of DL Software

There is no easy way to eliminate implementation-level nondeterminism. One direction is to enable the deterministic mode provided by DL libraries such as PyTorch, which avoids certain nondeterministic primitives provided by NVIDIA under the hood. However, implementing a deterministic mode is difficult and may not be possible for all DL functions, since the underlying NVIDIA primitives are closed source and their mechanism remains highly opaque.

Although training with deterministic mode can produce identical results across multiple FIT runs, many common APIs (e.g., 25 APIs in PyTorch) still lack deterministic implementation at the time of writing. For example, "MaxPool3d" and "NLLLoss" do not have a deterministic implementation in PyTorch [23]. Popular networks such as "Seq2Seq" [40] will not be trained if PyTorch is under deterministic mode.

**Overhead of deterministic mode**: The deterministic mode hurts the training speed. Our experiments show that the PyTorch deterministic mode adds (i) on average 17.8% of training-time overhead on FIT runs for technique S-Base, and (ii) 20.8% overhead training a ResNet101 and 14.1% overhead training a DenseNet121 with FIT runs.