# OpenReview forum: "Are My Deep Learning Systems Fair? An Empirical Study of Fixed-Seed Training"
_NeurIPS.cc/2021/Conference — NeurIPS 2021 Poster_

### Official Review · Reviewer_YRQu · 2021-07-12

**Rating:** 5
**Confidence:** 4

**Summary:**

In this paper, the authors investigate the impact of software implementation on the fairness variance in the training of DL models. They perform an empirical study of DL fairness with 27 techniques (22 mitigation techniques and five baselines) on four popular datasets (CelebA, MS-COCO, imSitu, and CIFAR-10S) and three DL networks (ResNet-18, ResNet-50, and NIFR), measured by seven popular bias metrics. Their results show that models considered fair for one specific run might not be fair in another fixed-seed identical training run. There are up to 12.6% fairness variance across identical training runs with identical seeds. Their literature survey shows that while fairness is gaining popularity in AI conferences, only 34.4% of the papers use multiple identical training runs to evaluate their approach.


**Limitations And Societal Impact:**

The authors adequately addressed the limitations and potential negative societal impact of their work.

**Main Review:**

# Strengths

. Target an important problem in deep learning.

. Extensive experiments.

# Weaknesses

- Novelty is a bit weak
- Limited scope of the literature survey

This paper describes an empirical study to quantify the impact of software implementation on the fairness and its variance of DL systems. It targets an important problem of Deep learning (DL) systems, which demand fairness. The paper finds that DL software implementations introduce fairness variance: identical DL training runs with a fixed seed produce different results. Such variance could make DL models and networks violate fairness compliance laws, resulting in negative social impact. The authors performed extensive experiments: they evaluated the fairness variance of 27 techniques using seven bias metrics (a total of 189 experiments). Statistical significance of our findings were also analysed.

The novelty of this work is a bit weak as the impact of software implementation on the variance of DL systems has already been described in some recent work.

For the software implementations, the paper did not consider the impact of various deep learning frameworks,
such as Tensorflow, Pytorch, MXNet, etc. Do different DL frameworks affect your findings? Furthermore, there are also many other dependent software packages such as Python, SciPy, etc.

The literature survey only covered 32 papers on DL systems’ fairness, from a few conferences (NeurIPS, ICML, AAAI, CVPR, ICCV, ECCV, FAccT). It is not clear if these papers are complete and representative enough.


**Time Spent Reviewing:**

3 hours

---

> ### Author Response · Authors · 2021-08-10
> **Thank you for the detailed insightful comments, we will make changes accordingly**
>
> **[Novelty compared with other work on variance from DL framework]**
>
> We did a thorough related work search and found that [51] is the only work that addresses the FIT variance of DL software implementations but doesn’t address fairness variance.
>
> R-T4tk puts it nicely “The result is surprising and although I knew that the same FIT runs can produce different outcomes, I didn’t know that the outcome could be that different!“ which is in line with [51]’s survey: over 80% of the respondents do not know or are unsure about the variance caused by DL software implementations. For the small percentage of respondents who know of FIT variance, they don’t know how much variance there is.
>
> Fairness variance caused by different random seeds or by train/test split has been investigated in previous work [D, E] but neither considers FIT variance. In fact, our discovery would affect both studies (e.g., their validity may be affected by the variance caused by software implementations) since [D] didn’t perform multiple FIT runs per split and [E] only built one model per random seed.
>
> **Question to the reviewer**: In the review, you mentioned that the variance of DL systems has already been described in recent work. Is the one we cited [51] what you refer to? If not, could you kindly let us know of such related papers? Thank you!
>
> **[Completeness and representativeness of literature survey]**
>
> For the literature survey, we picked all the papers that are relevant to our scope: fairness on DL networks from **top** AI and ML conferences (NeurIPS/NIPS, ICML, AAAI, CVPR, ICCV, ECCV, ICLR, FAccT from 2016 to 2020) (L261 in Section 4). The 32 papers are not a random sample, they are the **only papers** that are related to DL fairness from the 274 fairness papers filtered by a list of keywords (fair, bias, discriminate, aware, balance, disparity, opportunity, audit, gender, stereotype, race, amplification, ethic) from 24,186 papers (L264 in Section 4). Based on the reviewer's comment, we also checked another top conference, ICLR, but didn't find any paper relevant to DL fairness for the date range we chose (2016 to 2020). There was only one relevant paper [77] on fairness in ICLR 2021.
>
> Since DL software implementation could also be a software engineering (SE) issue. We also surveyed top SE conferences including FSE, ICSE, ASE, and ISSTA, with a total of 2,093 papers. However, none of the paper fits our scope so we omitted them due to space limitations.
>
> **[Impact of other DL frameworks]**
>
> Within a limited time, we re-implemented technique S-Base in Tensorflow and conducted 16 FIT runs. We found a MaxDiff of 1.8% measured by BA and a MaxDiff of 0.8% measured by DP, which shows that fairness variance exists in other DL framework.
>
> The reason that we only used PyTorch models in this paper is that all fairness techniques meeting our criteria are implemented in PyTorch. Our criteria include relevance to DL fairness and has a replication package. Starting from over 26,000 recent AI/ML and SE conference papers, 32 are relevant, and “Out of the 32 papers relevant to the fairness of DL systems, only four [32, 43, 70, 71] have a working replication package so that we can reproduce their results.” Reimplementing all 27 techniques in other DL frameworks is not only time-consuming but also error-prone (took us three days for one technique S-Base). One must make sure the reimplementation is as close as possible to the original PyTorch one. Since modern DL frameworks share many implementation concepts (e.g., parallelism and auto-tuning) as well as underlying libraries such as cuDNN and CUDA, they are likely to suffer from the same variance problem.
>
> References:
>
> [D] Friedler, S. A., Scheidegger, C., Venkatasubramanian, S., Choudhary, S., Hamilton, E. P., & Roth, D. A comparative study of fairness-enhancing interventions in machine learning. In FAT’19 (This is the Friedler et al. (2019)  suggested by R-fhvk).
>
> [E] Amir, S., van de Meent, J. W., & Wallace, B. C. (2021). On the Impact of Random Seeds on the Fairness of Clinical Classifiers. NAACL'21

---

> ### Author Response · Authors · 2021-08-31
> **Novelty over existing work on the variance of DL systems**
>
> Dear Reviewer YRQu,
>
> Thank you for your valuable reviews!
>
> In the review, you stated that the variance of DL systems has been described in other recent work. We conducted a thorough related work search and found no paper other than [51] addressing the FIT variance caused by software implementations. Our paper is still novel given the existence of [51], since [51] doesn’t study the fairness variance ([51] studied the variance of accuracy and time only).
>
> We would love to learn the details (e.g. references to existing work), hopefully before the initial decision window starting on [September 2nd](https://neurips.cc/Conferences/2021/Reviewer-Guidelines), so that we can improve our work and have a chance to explain the differences with existing work if any.
>
> Thank you very much for your time and information!

---

### Official Review · Reviewer_fhvk · 2021-07-16

**Rating:** 7
**Confidence:** 4

**Summary:**

This paper investigates the impact of so-called implementation-level nondeterminism of machine learning frameworks on state-of-the-art fairness methods. Unlike algorithmic nondeterminism, which is caused by, e.g., dropout or random shuffling and can be rendered deterministic by fixing the random seed, implementation-level nondeterminism is a consequence of parallelism and non-associativity of floating-point arithmetic. Thus, even fixed-seed training can lead to different results, and prior work has shown that the accuracy can vary by up to 2.9% across identical runs. This paper empirically demonstrates that implementation-level nondeterminism also impacts many fairness methods and can lead to variations of up to 12.6% in standard fairness metrics.

**Ethical Concerns:**

There are no ethical issues with this paper.

**Limitations And Societal Impact:**

The paper adequately addressed the limitations and potential negative societal impact.

**Main Review:**

**Originality:** The paper is the first to empirically show that implementation-level nondeterminism can negatively impact state-of-the-art fairness methods. These findings are different from those of prior work by Friedler et al. (2019), which this paper needs to cite, showing that fairness methods are sensitive to different training-test splits. The related work is generally cited adequately, but Finding 6 should indicate which 32 papers were considered.

**Quality:** The submission is technically sound. The paper uses various statistical tests to compute the significance of the results. However, the paper could improve the motivation for using these specific tests. For example, multiple tests for effect sizes based on the difference between means exist, but the paper only uses Cohen’s d. Moreover, the paper should repeat the experiments with different fixed seeds to see how the results are affected.

**Clarity:** The paper is generally clearly written and well organized. However, I think the causes of implementation-level nondeterminism should be moved from Appendix E to the introduction to provide a better overview of the problem setting. In addition, there are some mistakes, such as missing words (e.g., L087 most (83.8%) *researchers* are, L103 label *which* should, L105 even *though* the) and typos (e.g., L658 Google *Imgae* Search).

**Significance:** The results are meaningful to the fairness community as they elucidate the need for multiple fixed-seed runs when evaluating the merits of a novel method. However, as the paper mentions in Appendix E, this problem is also easily solvable using, e.g., PyTorch’s deterministic mode. In fact, the paper should conduct the same experiments using the deterministic mode for comparison. For this reason, I do not think that the current contribution is relevant enough to warrant publication at a top-tier conference.

**Questions**

- How does fixed-seed variance compare to the variance induced by varying the random seed?
- Did you run the experiments sequentially for roughly ten months, as indicated on L164, or does this represent the total running time?

**References**

Sorelle A. Friedler, Carlos Scheidegger, Suresh Venkatasubramanian, Sonam Choudhary. Evan P. Hamilton, and Derek Roth. A comparative study of fairness-enhancing interventions in machine learning. FAT 2019.

**Time Spent Reviewing:**

5

---

> ### Author Response · Authors · 2021-08-10
> **Thank you for the detailed insightful comments, we will make changes accordingly**
>
> **[Deterministic mode of DL libraries]**
>
> The deterministic mode does not solve the fairness variance problem for the following reasons. Thank you and we will clarify it in our paper.
>
> (1) People won’t use the deterministic mode if they don’t know the existence of fairness variance caused by DL software. According to the survey [51] of researchers and practitioners with DL experience, over 80% of the respondents do not know or are unsure about the variance caused by DL software implementations (let alone the deterministic mode of such DL software). Thus, the serious problems (such as compliance, research validity, and reproducibility) caused by DL fairness variance still remain. Publishing a paper at a top-tier conference is one of the best ways to raise awareness of this problem.
>
> (2) The deterministic mode doesn’t support all APIs or all libraries. Many (25) PyTorch APIs [A] do not have a deterministic implementation, including “MaxPool3d” and “NLLLoss”. Models such as the PyTorch implementation of Seq2Seq [B] will not be trained if PyTorch is in deterministic mode. TensorFlow’s deterministic mode is not deterministic, e.g., “tf.nn.sparse_softmax_cross_entropy_with_logits” is not deterministic [C]. We tested S-Base with this API in TensorFlow’s deterministic mode, and found a MaxDiff of 8.9% measured by the BA metric.
>
> (3) The deterministic mode hurts training speed. Our experiments show that the PyTorch deterministic mode adds (i) on average 17.8% of training-time overhead on FIT runs for technique S-Base, and (ii) 20.8% overhead training a ResNet101 and 14.1% overhead training a DenseNet121 with FIT runs.
>
> **[Impact of different seeds on the variance of FIT runs]**
>
> Within the limited time, we ran one additional experiment for S-Base with a different seed and found the result to be similar (a MaxDiff in normalized disparate impact of 2.1% vs. 2.0% in the paper). It would take about 24 months on four 2080Ti GPUs to repeat our experiments with 10 different seeds to obtain a statistically significant result. Thank you for pointing out the great idea of quantifying the impact of different seeds on the variance of FIT runs. We will investigate this direction as future work.
>
> **[Variance introduced by varying the random seed]**
>
> The submission focussed on quantifying the variance caused by FIT runs because it is less known than the variance introduced by varying the random seed (only 16.2% participants know variance from FIT runs vs 46.4% for variance from random seeds) according to the survey in [51]. Thank you for pointing this out. This remains as our future work.
>
> **[Total running time]**
>
> The 7,117 hours (~10 months) reported in L164 is the sequential running time for all our experiments. With 4 2080Ti, the wall clock time for all the experiments in our paper is roughly two and a half months.
>
> **[32 papers, citations, and writing issues]**
>
> We will add the list of 32 papers studied in the appendix, cite Friedler et al. (2019),  and fix all the language issues. Thank you for pointing them out!
>
> Reference:
>
> [A] https://pytorch.org/docs/stable/generated/torch.use_deterministic_algorithms.html#torch.use_deterministic_algorithms
>
> [B] https://github.com/IBM/pytorch-seq2seq/blob/f146087a9a271e9b50f46561e090324764b081fb/seq2seq/loss/loss.py#L104
>
> [C] https://github.com/tensorflow/tensorflow/issues/38185

---

### Official Review · Reviewer_T4tk · 2021-07-17

**Rating:** 7
**Confidence:** 3

**Summary:**

The paper did a thorough analysis of the effect of software implementation (e.g., floating-point and parallelism) on bias measures. It has been previously shown that FIT runs produce different accuracies! The result is surprising and although I knew that the same FIT runs can produce different outcomes, I didn’t know that the outcome could be that different! And it is a good study for the fairness community to be aware of!

**Main Review:**

The paper did a thorough analysis of the effect of software implementation (e.g., floating point and parallelism) on bias measures. It has been previously shown that FIT runs produce different accuracies! The result is surprising and although I knew that the same FIT runs can produce different outcomes, I didn’t know that the outcome could be that different! And it is a good study for the fairness community to be aware of!

The paper is very well written and empirical analysis is very thorough ranging from different biases, different architectures, etc.

Some suggestions (minor concerns):
I think it would be great if you report how much accuracies vary (from previous work, e.g., 51), and compare it with fairness measures! Are fairness measures more brittle and experience a wider range of change?
I think showing how much these measures vary with (1) different train/test split (2) different random seeds, and (3) due to software implementation, would be great (even over a few architectures to just get the sense of different components in variations.  I mean if (1) leads to much higher variation than (3) then studying (3) may not be very interesting.
You emphasis a lot on the fact that increasing fairness measures will lead to decrease accuracy which is a very well-known fact, I think emphasizing on increase in the variance of accuracy or variance of bias measure is more interesting.


**Time Spent Reviewing:**

3

---

> ### Author Response · Authors · 2021-08-10
> **Thank you for the detailed insightful comments, we will make changes accordingly**
>
> **[Accuracy variance for all the experiments]**
>
> We will add details about accuracy variance for all our experiments. Thank you! The submission talked more about fairness variance because fairness variance from software implementation has not been quantified and we show the difference can be up to 12.6% (versus 2.9% accuracy difference in [51]).
>
> **[Impact of different seeds on the variance of FIT runs]**
>
> Within the limited time, we ran one additional experiment for S-Base with a different seed and found the result to be similar (a MaxDiff in normalized disparate impact of 2.1% vs. 2.0% in the paper). It would take about 25 months on four 2080Ti GPUs to repeat our experiments with 10 different seeds to obtain a statistically significant result.
>
> Thank you for pointing out the great idea of quantifying the impact of different seeds on the variance of FIT runs. We will investigate this direction as future work.

---

### Decision · Program_Chairs · 2021-09-27

**Decision:**

Accept (Poster)

**Comment:**

This is a very empirical paper that documents an interesting and surprising fact: fixed-seed identical training runs (FIT runs) can have surprisingly different fairness characteristics.

I think this is an important fact that deserves to be known widely in the community and will likely be used to justify a lot of follow up work. The paper also includes a systematic literature survey to argue that this is both a little known fact and one that can affect the validity of some existing results.

That said, while the specific focus on fairness is novel, and I don't disagree with the authors that more people need to be aware of the variance that can be introduced by various sources of implementation-level non-determinism, I think what the paper claims as implications of these results are mostly things that people should know to do already. For example, the suggestion that people should use statistical tests to check the validity of any proposed improvements is a good one, but it is not one that follows from this research. Given that training algorithms are fundamentally stochastic, applying statistical tests to confirm claimed improvements should be a requirement even if implementation-level non-determinism were not an issue.